# On the Interplay of the DNA Replication Program and the Intra-S Phase Checkpoint Pathway

**DOI:** 10.3390/genes10020094

**Published:** 2019-01-29

**Authors:** Diletta Ciardo, Arach Goldar, Kathrin Marheineke

**Affiliations:** Institute for Integrative Biology of the Cell (I2BC), CEA, CNRS, Univ. Paris-Sud, Université Paris-Saclay, 91198 Gif-sur-Yvette CEDEX, France; diletta.ciardo@i2bc.paris-saclay.fr (D.C.); arach.goldar@i2bc.paris-saclay.fr (A.G.)

**Keywords:** DNA replication, replication checkpoint, intra-S phase checkpoint, ATR, Chk1, MBT, initiation rate, numerical, theoretical models

## Abstract

DNA replication in eukaryotes is achieved by the activation of multiple replication origins which needs to be precisely coordinated in space and time. This spatio-temporal replication program is regulated by many factors to maintain genome stability, which is frequently threatened through stresses of exogenous or endogenous origin. Intra-S phase checkpoints monitor the integrity of DNA synthesis and are activated when replication forks are stalled. Their activation leads to the stabilization of forks, to the delay of the replication program by the inhibition of late firing origins, and the delay of G2/M phase entry. In some cell cycles during early development these mechanisms are less efficient in order to allow rapid cell divisions. In this article, we will review our current knowledge of how the intra-S phase checkpoint regulates the replication program in budding yeast and metazoan models, including early embryos with rapid S phases. We sum up current models on how the checkpoint can inhibit origin firing in some genomic regions, but allow dormant origin activation in other regions. Finally, we discuss how numerical and theoretical models can be used to connect the multiple different actors into a global process and to extract general rules.

## 1. Introduction

Maintaining the informational and structural integrity of the genome is a necessity for an organism to survive and proliferate. During its cell cycle, any eukaryotic organism uses a large part of its time and resources to prepare for the phase of duplication of its genome (S phase). In eukaryotic organisms, genome replication begins at multiple locations called “replication origins” [1]. Activation (firing) of the replication origins is characterized by the local opening of the DNA double helix and the synthesis of two new strands of DNA. From each fired origin, two replication forks propagate at constant speed in opposite direction and duplicate the surrounding DNA (elongation). The collision between two convergent forks gives rise to a merging event called “termination”. The coordination between the firing, elongation, and termination events is required to ensure a complete duplication of the genome during S phase. Disruption of the orchestration of these events can irreversibly damage the genome and lead to cell cycle deregulation [2].

The regulation of the S phase in many eukaryotic organisms involves a spatio-temporal pattern of genome replication [3,4,5,6]. Two classes of origins have been defined: the origins that fire at the beginning of the S phase (early origins) and the origins that are triggered at the end of the S phase (late origins). In response to genotoxic stress during S phase, early origins are triggered normally, whereas the firing of late origins is generally inhibited [4,5,6,7]. The cellular processes that modify the firing program of the replication origins in the event of genotoxic stress have been extensively studied [8,9]. The general structure and effects of biochemical pathways that regulate genome duplication and ensure genome stability are conserved among eukaryotes. These regulatory mechanisms, called cell cycle checkpoints, are activated by the detection of an anomaly and target the main actors of the cell cycle in order to slow down the progression of the cell cycle until the problem is resolved [10]. In response to replication fork stalling, cells activate the intra-S phase checkpoint (also called the DNA replication checkpoint) [11,12]. DNA damage during S phase also activates the related, but separable, DNA damage checkpoint that shares some components with the replication checkpoint [13]. The DNA damage sensing kinase, ATR (Ataxia telangiectasia and Rad3 related), and its functional homologs in *Saccharomyces cerevisiae*, Mec1, and in *Schizosaccharomyces pombe*, Rad3, are recruited to stalled replication forks, and once activated, phosphorylate a number of proteins, including the downstream effector kinase Checkpoint kinase 1 (Chk1). The phosphorylation and ensuing activation of Chk1 or its functional homologs, Rad53 (*S. cerevisiae*) and CheckingCds1 (*Sc. pombe*), requires the activity of several other proteins. In mammalian cells and the *Xenopus* in vitro system, these proteins include Claspin, TopBP1, the Rad9-Hus1-Rad1 (9–1–1 complex), and the Rad17-Rfc2-5 complex. Once activated, the replication checkpoint kinases block cell cycle progression, downregulate late origin firing, stabilize stalled replication forks, and facilitate the restart of collapsed forks.

In eukaryotes during an unchallenged S phase, the rate of replication origin firing increases over time to reach a maximum and then decreases rapidly and vanishes before the end of S phase [14,15]. In a consensus manner, theoretical models picture the essence of this temporal variation as the temporal competition between passive replication of a replication origin by an incoming replication fork emanating from neighboring origins and the firing of the replication origin by a limiting trans-acting replication factor [16,17,18,19]. These studies assign a privileged place to replication forks as one of the key factors that can modulate the normal progression of S phase. Interestingly, experimental studies show that the integrity and stability of the replication forks during genotoxic stress is ensured by checkpoint proteins [20] and forks recover their normal activity once the stress is removed [21].

Several recent reviews discuss how many different factors contribute to intra-S phase checkpoint activation, and fork stability and restart [12,22,23,24,25]. In this review, we discuss the role of the intra-S phase checkpoint in the regulation of the replication program, mainly in budding yeast, in vertebrate cell lines, and the *Xenopus* in vitro system and early embryos, with a focus on the development of numerical models in order to better decipher origin activation in space and time.

## 2. Licensing and Activation of Replication Origins

Cell division requires that two exact copies of each chromosome are synthesized during S phase. Both copies must be distributed to each of the two daughter cells. DNA synthesis during S phase is therefore necessarily an extremely precise process.

During late mitosis and the G1 phase of the cell cycle, the origin recognition complex (ORC) and replication factors Cdc6 and Cdt1 load the minichromosome maintenance proteins 2–7 (MCM2-7), which form the core of the replicative helicase, as inactive head-to-head double hexamers (DHs) onto double-stranded DNA (dsDNA). This step is termed origin licensing or pre-replicative complex (preRC) formation. During S phase, several replication factors are assembled on these preRCs to form the pre-initiation complex (preIC) which is finally activated. This step is called replication origin firing [26]. Replication of eukaryotic chromosomes is initiated at multiple replication origins. The control of origin firing must be rigorous. First, each origin must fire only once during an S phase to avoid successive replications of the same chromosomal region. In addition, the cell must ensure the activation of a sufficient number of chromosomal origins so that each chromosome is completely replicated, in coordination with all other chromosomes. Regulated firing of the replication origins begins with the action of the two S phase protein kinases, Dbf4/Drf1-dependent kinase (DDK) and cyclin-dependent kinase (CDK), that convert a preRC into active replication forks. This conversion depends on the recruitment at the origins of the Cdc45-MCM2-7-GINS (CMG) complexes that, once activated, can unwind the dsDNA and then encircle and translocate along single-stranded DNA (ssDNA) with a 3′ to 5′ polarity on the leading strand. In budding yeast it is well established that CMG assembly requires Sld3, Sld7, Sld2, Polε, and Dpb11 proteins [27]. Additional proteins, which include MCM10 and Ctf4 (And-1 in humans), are then required for DNA unwinding and the recruitment of Polα [27,28,29].

The conversion of preRCs into active replication forks can occur throughout S phase. Thus, some origins are activated at the beginning of S phase and are thus called early, whereas others will fire later. The classification into early versus late origins is a simplification because origins fire in a continuous, staggered manner [6,30]. Some origins are replicated in a passive way and are called dormant [31].

Replication timing correlates with transcription, chromatin structure, and nuclear organization. Whether replication timing simply results from stochastic initiation at independent origins that fire with different probabilities, or reflects a more deterministic program controlled by central regulators and implying correlations between activation of neighboring origins, is a matter of current debate. Once origins fire and DNA replication commences, cells need to balance accuracy, speed, and the consumption and distribution of relevant resources such as nucleotides and replication factors, to complete replication in an efficient manner [32,33,34]. Recent data suggest that limiting levels of key initiation factors govern the time of origin firing in budding yeast [35]. In particular, Sld2, Sld3, Dpb11, and Dbf4 have been shown to be limiting for replication initiation. This suggests a model for sequential firing of origins whereby limiting firing factors are first recruited to the most accessible origins, promoting their early activation, and must be released from these and recruited to less and less accessible origins until complete genome duplication [32].

Replication origins have historically been difficult to identify, but DNA microarrays and next generation sequencing (NGS) techniques have allowed genome-wide studies [26]. Replication timing profiles were highly reproducible, but not resolutive enough to map individual origins, whereas origin maps were more resolutive, but less concordant between studies. In a cell population, origins fire in only a fraction of the chromosomal copies (termed origin efficiency) and over a broad window of time rather than at a precise time. Owing to this stochasticity, a locus can be replicated with some probability by forks originating from any of the origins and moving in both directions. Genomic methods only provide a cell-population average picture of genome replication. Single molecule methods such as DNA combing, a DNA fiber stretching technique, are required to visualize individual forks, to evaluate cell-to-cell heterogeneity, to reveal rare but important events such as fork slowing or stalling that are smoothed out by population averaging, and to study local correlations between firing of neighboring origins or movement of adjacent forks.

## 3. Intra-S Phase Checkpoint Activation and Response

### 3.1. Replicative Stresses

The faithful replication of the genome is frequently challenged through stresses of exogenous or endogenous origin. The sources of replicative stress (RS) are numerous and involve aberrant replication origin firing, replication-transcription collisions, difficult-to-replicate sequences such as DNA lesions, G-quadruplexes, or microsatellites, depletion of deoxyribonucleotides (dNTP) pools, dysfunction of any of the numerous factors required for faithful replication, and origin underusage or origin overusage. Experimental induction of RS can be achieved, for example, by stalling DNA polymerases by reducing dNTP levels using hydroxyurea (HU) or by chemical polymerase inhibition by aphidicolin.Replication stress has various repercussions in the cell, but it involves a transient slowing or stalling of replication fork progression and/or DNA synthesis [12,36]. Many of the common markers used to detect replication stress reflect activation of the ATR pathway. ATR-dependent phosphorylation of RPA (Ser 33) or Chk1 (Ser 345), or detection of ssDNA either directly, through native bromodeoxyuridine (BrdU) immuno-fluorescence, or indirectly, through the formation of RPA foci, are indications of RS. However, the use of these RS markers assumes that all RS activates ATR to a high enough level to induce widespread phosphorylation of its downstream targets, or that all RS generates detectable patches of ssDNA, neither of which is necessarily true. For example, the cell may experience RS at one or a few stalled forks such that it responds locally, but not globally, to that stress. We therefore believe that the clearest read-out of RS is the direct measurement of replication origin firing and replication fork progression by sensitive methods like DNA fiber stretching techniques (DNA combing). These techniques allow visualization of the replication process at the level of single DNA fibers, but their throughput is low if one needs to map the genomic position of the observed fiber. However, recent advances in the study of the DNA replication process using nanopore sequencing [37,38] could soon allow direct detection of stalled forks on genomic positions on single DNA fibers with high throughput.

### 3.2. Intra-S Phase Checkpoint Activation and Fork Stability

The progression of replication forks may be compromised under certain conditions. To progress on the genome, replicative helicases must separate each downstream DNA base pair and progress even in the presence of DNA bound proteins or DNA structures that can act as barriers to easy unwinding of dsDNA. The DNA polymerases must incorporate nucleotides during synthesis with great fidelity while being in contact with the replicative helicases. To avoid unreplicated regions, all forks must be stable enough to be able to merge with a converging fork emanating from nearby origins. When a replication fork is blocked in its progression, the protein complex associated with it must remain stable and not disintegrate in order to resume synthesis once the obstacle has been overcome. The blocked replication forks trigger the intra-S phase checkpoint via a kinase activation cascade involved in the cellular response [31,36,39]. First Mec1/ATR kinase (budding yeast and metazoans, respectively) activates the second kinase, Rad53 in *S. cerevisiae* and Chk1 in metazoans. Moreover, the activation of the checkpoint in response to the blockage of the replication forks requires the Mrc1/Claspin mediating protein (*S. cerevisiae*/metazoans, respectively) which is present at the forks, even in the absence of replication stress [40]. Therefore, if stalled forks cannot restart, replication must be completed through incoming neighboring forks. When two convergent forks are irreversibly stalled, replication can only be rescued by the activation of “back-up” or dormant origins within the unreplicated gap [2,41,42,43,44]. Dormant origins fire close to stalled or slowed forks in mammalian cells [42,45,46] after HU treatment and in the *Xenopus* in vitro system [43] after aphidicolin treatment. In budding yeast, dormant origins are very late replicating origins and are thereby normally passively replicated from neighboring early origins [47,48].

A sufficient redundancy of potential origins licensed in G1 phase is a clear prerequisite for this mechanism to efficiently rescue replication. Indeed, during licensing a single ORC can load multiple MCM2-7 DHs onto dsDNA, and even though each MCM2-7 DH is competent to fire, only a small fraction of them actually do so during an unperturbed S phase. Unfired MCM2-7 DHs therefore provide dormant origins that can facilitate completion of normal S phase, or rescue artificially stalled forks. Thus, the loading of an excess of MCM2-7 DHs creates a redundancy of potential origins that helps to safeguard replication. As cells complete S phase, however, unfired MCM2-7 DHs are cleared from chromatin by still elusive mechanisms. MCM2-7 DHs cannot be reloaded on chromatin until the next G1 phase because of multiple cell cycle regulatory mechanisms which normally prevent DNA over-replication in a single cell cycle [49].

As blockage of replication fork progression is lifted, resuming synthesis and completion of chromosome replication requires the presence of functional checkpoint proteins. In the absence of checkpoint kinases, genome replication remains incomplete after removal of the blocking conditions due to the failure of the replication forks to resume synthesis. These forks unable to resume synthesis in the absence of a checkpoint are said to have “collapsed” because of the dissociation of certain proteins of the replicative complex and, in particular, the DNA polymerases seem to “fall out” of the forks [50,51]. It is therefore possible that the checkpoint proteins stabilize the replicative complexes blocked in their progression by directly phosphorylating one or more proteins of these complexes. Several proteins such as MCM2-7 and RPA are indeed phosphorylated by Mec1/ATR although the biological significance of these phosphorylations is not yet well understood.

### 3.3. Inhibition of the Replication Program by the Intra-S Phase Checkpoint

The first evidence that DNA replication is actively inhibited by the DNA damage checkpoint came from observations in AT (Ataxia telangiectasia) cells treated with ionizing irradiation [52]. Cells with mutations in *ATM* show radio resistant DNA synthesis (RDS) which was proposed to be due to less inhibition of replication initiation using alkaline sucrose gradients profiles. Later on in budding yeast, it was observed that S phase progression in Mec1 and Rad53 mutants is similar in control cells and cells treated with the alkylating agent MMS [53]. After the identification of specific origin sequences in budding yeast, it was shown that the Mec1/Rad53 dependent checkpoint inhibits or delays late origin firing in S phase in cells treated with hydroxyurea [54] and MMS [55]. It was thus proposed that the checkpoint serves to prevent a subset of origins from activation and to prevent the destabilization of active forks. Interestingly, the Mec1/Rad53 pathway may also regulate the timing of origin firing in the absence of damaging agents as one late origin fired earlier and one inefficient origin fired more efficiently in a Rad53 mutant [55]. The role of the intra-S phase checkpoint on the regulation of the replication program was also confirmed in later genome wide analyses of ssDNA formation in HU-treated wild-type (WT) and Rad53 mutant yeast cells [56]. ssDNA formation occurs at only a subset of origins in wild-type cells, but at virtually every origin in Rad53 mutant cells, suggesting that Rad53 prevents a subset of origins ("Rad53-checked origins") from being activated in the presence of hydroxyurea. Whether the activation of origins under the control of Rad53 reflects replication timing or activation efficiency or both was not clear. However, it was shown that the checkpoint in the presence of HU does not completely inhibit late origin firing, but only generally slows down S phase, which suggests that the checkpoint modulates S phase but not the origin timing program [57,58]. A Rad53 checkpoint deficient mutant massively de-represses late origin activation, whereas other partial checkpoint mutants allow less late origins to be activated [59], indicating that the inhibition of the replication program by the checkpoint is fine-tuned.

In mammalian or avian cell lines and in the *Xenopus* in vitro system, initiation of DNA replication was shown to be regulated by the intra-S phase checkpoint at different levels: single replication origins, clusters, domains, and replication foci. In mammalian cells, origins are inhibited at the ribosomal DNA (rDNA) locus in cells proficient for the checkpoint pathway after ionizing radiation, but not in AT cells [60]. The intra-S checkpoint inhibits the DNA synthesis in late replication foci in the presence of the replication inhibitor aphidicolin [46,61]. In asynchronous human and avian cells, ATR/ATM inhibition by caffeine, or Chk1 inhibition by UCN-01 or by Chk1 depletion, led to an increased origin density inside already activated clusters in all tested cell lines, and an increase in foci number only in avian cell lines [62]. In human cells, genome wide origin mapping on coding sequences by NGS showed that upon fork stalling in HU, only early origins fire in early domains, but upon checkpoint inhibition with the ATM/ATR inhibitor caffeine, origins fire in mid-to-late regions [63]. In the embryonic *Xenopus* in vitro system using single DNA fiber analysis, it was shown that ATR regulates origin firing during an unchallenged S phase [64,65], but no effect of Chk1 depletion or inhibition was detected in the *Xenopus* in vitro system [66,67]. Later, two other studies clearly showed that Chk1 depletion or inhibition in the *Xenopus* in vitro system resulted in increased overall origin firing [68,69], demonstrating the role of Chk1 in the negative regulation of DNA replication in the absence of exogenous replication stress. Upon Chk1 or ATR inhibition an important increase of global fork densities was found, but no difference in local origin distances was observed after Chk1 inhibition and only a modest decrease was seen upon caffeine treatment [64,69]. These two apparently contradictory results suggest that the checkpoint mainly inhibits origins at the level of later replicating clusters in the *Xenopus* in vitro system, but less inside already activated clusters, in contrast to mammalian cell lines. This difference between differentiated cells and the non-differentiated embryonic system might be linked to the overall higher fork density and therefore less available dormant origins in the shorter embryonic S phase.

These different observations illustrate that the checkpoint inhibits origin firing at different chromosomal organization levels and often in inactive, late clusters or domains, depending on the organisms. On the other hand, some experiments showed an increased origin firing due to dormant origin activation when forks are slowed down which allowed local rescue of replication. In order to explain how the checkpoint in response to DNA damage can simultaneously inhibit and promote origin firing, different models have been proposed [70] which could also be valid for the regulation in response to endogenous or exogenous replication stress (Figure 1).

In the absence of replication stress, origins in early domains are activated in the beginning of S phase (on the time scale t_0_–t_1_) before origins in late domains at consecutive times (t_2_–t_3_) later in S phase (Figure 1a). In the presence of replication stress, the activation of origins in late domains is inhibited by the intra-S phase checkpoint (ISC) (Figure 1b–d). Since the inhibitory checkpoint signal is produced in early domains where replication forks are stalled, it should normally inhibit origins close to stalled forks before it is transmitted to late domains. There could be at least three, not mutually exclusive, possibilities to explain dormant origin firing in these early domains. First, dormant origins can fire because they have already assembled functional preICs, due to only a slight asynchrony in the assembly of the preICs between early and dormant origins (Figure 1b). Thus, the inhibitory step of the checkpoint in the activation cascade of origins has been passed. In a second possibility, there could be a local inhibition of the inhibitory action of the checkpoint close to stalled forks inside early domains, resulting in the activation of dormant origins (Figure 1c). This unknown mechanism (X) could involve for example Plk1, recruited in a checkpoint dependent manner via MCM2-7 to stalled forks and leading to a decrease of Chk1 activity [71]. It could also implicate other checkpoint recovery mechanisms like phosphatases directly inhibiting Chk1 (see chapter 4). Third, the checkpoint could directly activate dormant origins inside early clusters by targeting proteins that are distinct from those responsible for late origin inhibition. Future studies will help to distinguish between these scenarios or reveal others.

### 3.4. Molecular Mechanisms of Checkpoint Dependent Origin Inhibition

#### 3.4.1. In Response to Exogenous Stress

The intra-S phase checkpoint inhibits late origins by preventing the conversion of the preRC into a functional preIC. This conversion is dependent on the S phase kinase complexes, CDK and DDK (Figure 2). The sequential assembly of the preIC has been deciphered in budding yeast, but the order of events is less clear in higher eukaryotes. DDK acts before CDK in budding yeast [27] and in the *Xenopus* in vitro system [72,73]. DDK phosphorylates several MCM subunits [74,75]. A recent study suggests that phosphorylation of the N-terminal tails of MCM4 and 6 could serve as a landing platform for Sld3 (associated to Sld7) and allow Cdc45 recruitment [76]. Sld2 and Sld3 are the essential CDK substrates in budding yeast [77,78]. CDK-dependent phosphorylation of Sld2 and Sld3 leads to their binding to Dpb11 and the recruitment of GINS and Polε. In vertebrate cells, phosphorylation of the Sld3 ortholog Treslin/TICCR promotes the binding of TopBP1, ortholog of Dpb11 [79,80,81], with RecQL4 being the ortholog of Sld2 [82,83]. DDK and CDK are both targets of the intra-S phase checkpoint, but apparently not in the same way in budding yeast and vertebrates.

The main effector checkpoint kinase Rad53/Chk1 is phosphorylated and activated by Mec1/ATR, dissociates from chromatin, and phosphorylates several down-stream effectors. The best characterized target of Chk1 is phosphatase Cdc25, which is necessary for the activation of Cdk1 and Cdk2 through the dephosphorylation of Tyr15 and Thr14. Vertebrate cells have three isoforms: Cdc25A, B, and C. Due to their phosphatase activity, Cdc25B and Cdc25C are primarily responsible for the G2/M transition, while Cdc25A controls G1/S and G2/M transitions and progression through S phase [84,85]. In human cells, after UV irradiation, the 14-3-3γ protein binds to Chk1 autophosphorylated at S296 to promote Cdc25A phosphorylation at S76 [86]. The Cdc25A phosphorylation at S76 upon different types of stress (UV, IR irradiation, HU) results in βTrCP-dependent Cdc25A degradation [87,88,89,90,91,92]. However, it has been shown that the intra-S phase checkpoint remains active after IR, UV radiation, or osmotic stress even in the presence of a nonphosphorylatable and undegradable Cdc25A (S76A) protein [93]. This might be because in the *Xenopus* in vitro system and human cells Chk1 additionally phosphorylates Cdc25A on Thr504 (human Thr507), thereby directly inhibiting the interaction of Cdc25A with Cdk1 and Cdk2. This inhibition seems more important than degradation of Cdc25A for the intra-S phase checkpoint in early *Xenopus* embryos [94,95]. In budding yeast, Cdk activity is essential to avoid the re-loading of MCM2-7 onto origins that have already fired. Rather than inhibiting CDK directly, Rad53 phosphorylates Sld3 after induced replication stress by HU. This phosphorylation prevents the interaction of Sld3 with Dpb11 and Cdc45 and inhibits late origin firing [96,97]. A more detailed view was provided in a recent study showing that Rad53 can create its own docking site on Cdc45 by phosphorylating it. The docking of Rad53 on phospho-Cdc45 then leads to the phosphorylation of Sld3 [98]. Contrary to yeast, in human cells the phosphorylation of Treslin, ortholog of Sld3, is reduced after treatments with HU. The hypophosphorylation is Chk1-dependent and reduces the binding of Treslin to TopBP1, human ortholog of Dpb11, but it could be indirectly due to the inhibition of Cdk activity by Chk1 [81]. 

In budding yeast, the intra-S phase checkpoint also targets DDK. Dbf4 is phosphorylated by Rad53 upon HU and MMS treatment. After phosphorylation, Dbf4 dissociates from chromatin [99], DDK activity decreases [100,101], and origin firing is inhibited [96,97]. Sld3 and Dbf4 seem to regulate late origin firing after HU through independent, but overlapping, pathways [102]. In *Xenopus* and human cells, the results on how the checkpoint inhibits DDK activity are contradictory. In the *Xenopus* in vitro system, it has been shown that etoposide treatment inhibits DNA replication by decreasing Cdc7 activity, Dbf4 binding to chromatin, and the stability of the Dbf4-Cdc7 complex; normal conditions can be restored through inhibition of ATR by caffeine or neutralizing ATR antibodies [103]. In response to UV-C or ionizing irradiation, the ATR/Chk1 dependent DNA damage pathways also inhibits Cdc7/Dbf4 [104,105]. Although other studies agree on the role of DDK in DNA replication inhibition after replication stress, they show that the etoposide-dependent release of Cdc7 from chromatin is not restored by caffeine [106], and that etoposide does not perturb complex formation, chromatin association, and kinase activity of Dbf4-Cdc7 and Drf1-Cdc7 [107,108]. It was therefore proposed that DDK plays an active role in checkpoint regulation by targeting either the checkpoint activation or the recovery pathways [107]. In another study, there is evidence that the checkpoint seems to act directly on the dephosphorylation of the DDK substrate MCM4 through the recruitment of PP1 to chromatin [106]. In humans cells, Dbf4-Cdc7 and Drf1-Cdc7 complex formation, chromatin association, and kinase activity are not perturbed after HU or etoposide treatment [109]. The stabilization of the Dbf4-Cdc7 complex upon HU treatment is due to Chk1-dependent degradation of Cdh1, that prevents degradation of DDK by Anaphase-promoting complex (APC)/C [110]. Moreover, ATR and ATM directly phosphorylate Dbf4 on chromatin. The phosphorylation suppresses DNA replication, but does not reduce the activity and stability of the Dbf4-Cdc7 complex [111]. A possible reason for preserving DDK activity after replication stress is that it could be required to maintain fork stability by promoting translesion synthesis (TLS) [112].

Another mechanism by which the intra-S phase checkpoint blocks helicase activity independently of CDK and DDK inhibition in mammalian cells is through ATR-dependent phosphorylation of the histone methyltransferase in myeloid/lymphoid or mixed-lineage leukemia (MLL) [113]. This leads to MLL stabilization and its accumulation on chromatin, where it methylates histone H3 Lys4. This modification prevents loading of Cdc45 at nearby replication origins, and thus suppresses origin firing.

The Fanconi anemia (FA) pathway is known to repair DNA interstrand cross-links during S phase by orchestrating the interplay between multiple DNA repair pathways [114,115]. In response to low replication stress (HU) and in the absence of FANCD2 and FANCI, core components of the FA complex, origin firing is altered. FANCD2 and FANCI interact with the MCM2-7 complex at the replisome [116] and are involved in the regulation of dormant origin firing [117,118,119].

#### 3.4.2. In Response to Endogenous Stress

The intra-S phase checkpoint also regulates the activation of origins in the absence of exogenous replication stress. In the *Xenopus* in vitro system, ATR/ATM inhibition by caffeine increases the rate of DNA synthesis by increasing the number of fired origins in early S phase [43,64,65].

Different observations show that inhibition or depletion of Chk1 in human and chicken cells increases DNA replication due to a higher number of fired origins. This was confirmed by the increased Cdc45 recruitment in the absence of Chk1 [62,120,121,122]. In the same conditions, a decrease in the replication fork speeds was observed [123] that is at least in part a consequence of the increased number of fired origins [124]. Little is known about the mechanism of action of the intra-S phase checkpoint during an unperturbed S phase. Chk1 is regulating Cdk activity during normal cell cycles [39,69]. In particular, in human cells, Chk1 phosphorylates Cdc25A at different sites to destabilize it and loss of Chk1 leads to an increase in Cdc25A levels [93,125,126,127]. Chk1 also inhibits origin firing through direct interaction with Treslin in *Xenopus* and humans [68], and phosphorylates MCM3 at S205 in human cells to inhibit DNA replication [128]. Even if a direct action on DDK has not been proven in human cells, Chk1 prevents the recruitment of And-1 to GINS in a Cdc7-dependent manner. Moreover, the inhibition of Chk1 induces hyperphosphorylation of MCM4, a substrate of DDK [129]. The role of the intra-S phase checkpoint during a normal S phase could be to adapt the rate of origin firing to the availability of limiting replication factors at active replication forks [64], in addition to the response to the presence of endogenous replication stress.

In budding yeast, checkpoint kinase Rad53 does not appear to control late origin firing through direct phosphorylation of Sld3 and Dbf4 in a normal S phase [102], but late origins fire earlier in checkpoint mutants [55].

Since the intra-S phase checkpoint in normal S phase seems to regulate the same proteins that are targeted in the presence of replication stress, more studies are required to assess if the pathways in the presence or absence of exogenous replication stress are different or if they are just amplified to face anomalies of the process.

## 4. Intra-S Phase Checkpoint Recovery and Adaptation

An important task of the checkpoint is to allow time for the restart of forks or repair of DNA damage, so that cells can resume cell cycle progression. This is an active process and is called checkpoint recovery when DNA damage is repaired, but checkpoint adaptation when the cell cycle continues in the presence of unrepaired damage or persistent fork stalling. As the checkpoint also senses perturbations when replicating repetitive sequences and fragile sites, or when replication factors are limiting, it is likely that recovery or adaptation from checkpoint activation also takes place in the absence of exogenous replication stress. During intra-S phase checkpoint recovery, checkpoint signaling needs to be inactivated before resuming DNA replication by the restart of replication forks. During recovery after DNA double strand breaks or DNA damage, Rad53 is dephosphorylated and inactivated by the PP2C-like phosphatases in budding yeast [130,131], and Chk1 by PP2A [132] and Chk2 by WIP1 in human cells [133]. But in the absence of PP2C-like phosphatases fork restart is delayed, whereas DNA replication can be completed by late origin firing [134]. After replication stress, different phosphatases might therefore be necessary to inactivate Rad53 in budding yeast [135]. Another mechanism is to target Rad53/Chk1 inactivation via the adaptor protein Mrc1/Claspin. In the *Xenopus* in vitro system, during adaptation upon replication fork stalling in the prolonged presence of aphidicolin, ATR phosphorylates Claspin that creates a docking site for Plk1. Plk1 then phosphorylates Claspin which leads to its dissociation from chromatin and the subsequent inactivation of Chk1 which abrogates the checkpoint and restores progression into G2/M [136]. In human cells, Claspin phosphorylation by Plk1 induces its degradation by the proteasome after targeting by the SCFβ^TrCP^ ubiquitin ligase [137,138,139,140]. Another study showed that Plk1 is recruited to stalled forks induced by aphidicolin dependent on phosphorylated MCM2 by ATR, and Plk1 depletion leads to increased levels of phospho-Chk1 in the *Xenopus* in vitro system [71]. Checkpoint signaling can also be attenuated by alterations in chromatin structure and checkpoint protein complex architecture [141].

## 5. DNA Replication and Intra-S Phase Checkpoint Response in Rapid Cell Cycles During Early Development

Rapidly dividing cells treated with DNA damaging agents, aphidicolin, hydroxyurea, or UV, continue to divide in early embryos of *Xenopus* [142,143,144], *Drosophila* [145], Zebrafish [146], and *C. elegans* [147]. This led to the conclusion that DNA damage and intra-S phase checkpoints are absent in early development stages and become activated only later during development when checkpoint proteins become essential. However, all checkpoint pathways are functional in the *Xenopus* in vitro system which mimics very early *Xenopus* embryos. How can these different observations be reconciled to resolve this apparent contradiction?

After fertilization of the egg, a series of rapid and synchronous cell division cycles is initiated in most animal species [148]. In these cleavage cycles, S phase and M phase alternate without intervening gap phases (G1 and G2 phase). Little or no cell growth occurs during these cell cycles due to the absence of transcription. After 12–13 cell divisions in *Xenopus*, a threshold ratio of DNA to cytoplasm is reached, which leads to cell cycle slowing [149,150] and resumption of zygotic transcription. This event is called the mid-blastula transition (MBT), after which the length of the cell cycle increases [151,152].

The extremely short S phase of only few minutes in the cleavage cycles requires a rapid but faithful duplication of the genome in comparison to cells at later stages of development and differentiation with S phases lasting several hours. In post-MBT embryos, initiations become restricted to the intergenic regions at the rDNA locus, whereas initiations are allowed everywhere in pre-MBT embryos at this locus [153]. Increase of the DNA to cytoplasm ratio in the *Xenopus* in vitro system, which mimics the increase of the ratio of DNA to cytoplasm in vivo, leads to an increase of S phase length [154]. DNA combing analysis showed that the activation time of different replication clusters is spread out, leading to the increase of S phase length [64]. Altogether these observations led to the conclusion that the replication program changes at the MBT [155].

Several lines of evidence suggested that titration of a maternally deposited factor by nuclear components triggers the activation of the intra-S phase checkpoint and thereby the change of the replication program at the MBT. Pre-MBT *Xenopus* embryos exposed to high concentrations of aphidicolin or hydroxyurea continue to divide despite incomplete replication before the MBT [142] which illustrates the absence of the G2/M phase checkpoint. But the checkpoint can be activated in pre-MBT embryos by the injection of a threshold concentration of damaged and undamaged DNA, suggesting that all components of a functional DNA damage checkpoint are present in pre-MBT embryos [156,157]. Downregulation of Xenopus Chk1 by microinjection of a dominant negative form showed that XChk1 is vital for cell cycle remodeling after the MBT even in the absence of exogenous stress [158], as was shown for the Chk1 homologue in *Drosophila* [159]. Chk1 becomes transiently activated at the MBT [158] and induces the developmental degradation of Cdc25A [158] and Drf1 [160] necessary for cell cycle lengthening at this stage. Recently, it was shown that overexpression of four replication initiation factors (Treslin, Drf1, TopBP1, RecQ4) delays the onset of the MBT in *Xenopus laevis* embryos and reduces replication track distances [161]. Altogether, these different observations support the model in which limiting maternal DNA replication factors are titrated during late cleavage stages, leading to an increase in S phase length and a replication-checkpoint dependent cell cycle delay. However, upon overexpression of these replication factors, the transient Chk1 phosphorylation stays high in MBT embryos, suggesting that other additional limiting factors or other mechanisms are necessary for the transient developmental activation of Chk1 at the MBT. In addition, other non-limiting factors such as non-coding Y RNAs [162], the chromatin remodeling complex xNuRD [163], histones [164], and Rap1 interacting factor (Rif1) [165] have been reported to be linked to DNA replication program changes at the MBT.

Some observations challenge the idea that a critical threshold of stalled forks is needed to activate the ATR/Chk1-dependent intra-S phase checkpoint when the DNA to cytoplasm ratio increases both in vitro and in vivo (Figure 3). In the in vitro system, Chk1 phosphorylation incrementally increases with increasing DNA to cytoplasm ratios. Upon treatment with aphidicolin, Chk1 phosphorylation does not increase at high DNA to cytoplasm ratios in comparison to low DNA to cytoplasm ratios when the phospho-Chk1 signal is normalized to the number of active forks [67]. In addition, at low DNA to cytoplasm ratios, late origin firing is inhibited in the presence of aphidicolin in vitro and Chk1 phosphorylation in pre-MBT embryos treated with aphidicolin can be detected [69]. In a recent study, direct microinjection of aphidicolin into dividing embryos provoked a robust cell cycle slow down or arrest [160]. Finally, in the absence of exogenous stress, ATR/ATM inhibition, or Chk1 depletion or inhibition, the checkpoint is de-repressed which leads to increased origin firing [64,65,68,69].

It is possible that the intra-S phase checkpoint in early embryos induces a gradual response to stalled forks rather than an all or nothing response, but either the downstream signal preventing the entry into M phase is too low, or cell cycle progression is allowed despite persisting replication stress. One possible explanation would be that mechanisms that inhibit or bypass the checkpoint pathways are more active in pre-MBT embryos. Protein phosphatase PP2A, as part of the checkpoint recovery pathway, antagonizes ATR/ATM [167] and PP2A becomes limiting for DNA replication rate upon increase of the DNA to cytoplasm ratio [67] in the *Xenopus* in vitro system, however, the targets of PP2A in this context have not been identified. In *C. elegans*, mutations in the E3 Sumo-ligase gei-17 or the TLS (translesion synthesis) polymerase polη sensitizes early embryos to the DNA damaging agents MMS or UV but not to HU [147]. In *Xenopus*, it was shown that RAD18 ubiquitin ligase, a DNA damage tolerance protein, is abundant in early pre-MBT embryos and declines before the MBT, whereas overexpression of a RAD18 mutation sensitizes pre-MBT embryos to UV irradiation [168], but did not affect embryos in the absence of UV. The latter mechanism does not interfere with the developmental activation of the intra-S phase checkpoint as seen by the transient phosphorylation of Chk1 at the MBT. Finally, work in *Drosophila* has suggested that the conflict between replication forks and early zygotic transcription is a trigger for intra-S phase checkpoint activation [169], but it is unknown whether a similar mechanism exists in *Xenopus*. Therefore, it is probable that several different mechanisms contribute to the activation of the intra-S phase checkpoint and modulation of the DNA replication program during early development.

## 6. Numerical Simulations and Models of the Spatio-Temporal Program of DNA Replication

As described above, the regulation of the DNA replication process is the sum of the actions of many different molecular factors. Our understanding of the spatio-temporal replication program continuously progresses thanks to technical advances that have enhanced the quality and quantity of available experimental data. Nowadays, it is commonly accepted that stochasticity plays an important role in the replication process [34]. The intrinsic temporal variability and the large number of molecular factors involved in this process imply that the temporal dynamics of a replicating cell population cannot be apprehended by studying only parameters defining the firing of a single replication origin, or only interactions between a single replication factor and replication origins. Numerical and theoretical models are essential for the analysis and comprehension of this complex process. Indeed, mathematical models allow picturing the spatio-temporal properties of a replicating cell population as an emerging phenomenon resulting from the spatio-temporal correlations of the interactions among actors involved in the DNA replication process.

### 6.1. Different Approaches to Analyze the Spatio-Temporal Replication Program

The rate of origin firing was first measured using data from DNA combing as the number of newly formed replication forks of size 3–8 kb in the *Xenopus* in vitro system [30]. Later on, Bechhoefer and colleagues used these data to construct the rate of origin firing, I(t), per unit of time and per length of unreplicated DNA [170]. They used I(t) as an input in a one-dimensional nucleation-and-growth formalism (KJMA) to quantitatively model the dynamics of parameters that govern the DNA replication program in the *Xenopus* in vitro system. In the KJMA framework, initiations are supposed to be uncorrelated and homogeneously distributed along the genome. The I(t), extracted from DNA combing data, showed that origin activation takes place throughout S phase with a probability increasing for most parts of S phase before decreasing at the end. This final decrease was considered as an experimental artifact and was not included in the analysis [171,172].

Later on, in an effort to explain the biological significance of an increasing initiation rate, Yang and colleagues [173] found that the obtained I(t) minimizes the consumption of resources while assuring a robust control of the typical replication time. Moreover, a strikingly similar temporal profile of the replication initiation rate was found in different eukaryotes: *S. cerevisiae*, *S. pombe*, *D. melanogaster*, *X. laevis*, and *H. sapiens* [14]. In order to find a general mechanism that could explain the temporal profile of experimentally measured I(t), numerical simulations were used [15]. The measured I(t) was reproduced by considering that an origin is fired through the encounter with a limiting replication factor, that the probability of encounter is modulated with fork-density in a self-limiting manner, and the amount of the limiting-factor linearly increases during S phase. In another work [19], the initiation rate was defined by the time that a “searcher” protein needs to find a potential origin through a 3D–1D diffusion process. They found that the increasing part of I(t) is defined by a reaction-limited regime, while at the end of S phase (the decreasing part of I(t)), it is defined by a diffusion-limited regime.

More recent simulations reproduced I(t) without the need of specific assumptions to describe the final decreasing part. In [17], the initiation process in a cell population was described as a scattering phenomena in an inhomogeneous medium. I(t) was reproduced by taking into account the conformation of the genome and the diffusion of replication limiting-factors inside the nucleus. In [16], the temporal program was depicted by considering a limited number of potential origins and a limiting replication factor that increases early in S phase until reaching a constant value. The time needed to replicate the DNA between two adjacent potential origins can alone account for the final decrease in I(t). The study of I(t) alone has brought some advances in the comprehension of the dynamics of origin firing. However, to have a global vision of the replication process, it is necessary to take into account the local modulation of I(t) and particularly to assess if there exists a spatial correlation between initiation events both at the level of the linear genome and/or at the level of the three-dimensional conformation of the genome inside the nucleus.

Using fitting or mathematical inversion procedures the local rate of initiation, I(x,t), was extracted from yeast replication timing data [174,175,176,177,178]. These studies emphasized that late origins are less efficient in their firing than early ones, and that the local rate of initiation I(x,t) at an origin position increases throughout S phase. Yang and colleagues [174] extracted the intrinsic cumulative firing time distribution for each origin and found that there is no correlation between the median firing time of consecutive origins. They conclude that origins continuously fire throughout S phase, so that it is not possible to distinguish clusters of origins activated synchronously early or late in S phase. However, this interpretation could be questioned because it is founded on the KJMA framework where the independence of origin firing is one of the assumptions of this model. Indeed, in fission yeast, a clear 3D spatial segregation of replication origins according to their firing times and efficiency was demonstrated [179], and the temporal program of origin firing was proposed to be stochastic, but coordinated by the local chromatin architecture in conjunction with the diffusion of a mobile firing factor from the spindle pole body (SBP). Furthermore, Ma and colleagues [180] used a model in [15] to analyze combing data from budding yeast and showed a clear correlation between I(t) and replication fork density. This implies that origin firing is connected to the presence of an already activated fork.

In mammalian cells, different models hypothesize a linear clustering of origin firing. One of the models proposes that the presence of a replication fork can increase the firing rate of nearby origins. For example, in order to describe the S phase program by modeling features of chromatin structures, a “next-in-line” model has been used [181]. In this model, the chromatin environment defines the sites that are selected for initiation of DNA synthesis, but then replication spreads from these primary initiation sites through the replication fork that increases local firing probabilities. In [182], by using DNA combing data, genome-wide replication timing data, and origin mapping data, a “domino model” was proposed for origin activation in which replication forks progressing from early origins stimulate initiation in nearby unreplicated DNA. More recently, this domino-like model was combined with an inhibition of firing at distances below the size of chromatin loops to reproduce the empirical properties of DNA replication in human cells [183].

While it seems clear from these studies that at least in higher eukaryotes the activation events are not homogeneously distributed along the genome, more efforts are necessary to understand if they are correlated in space and time and what kind of mechanism could be responsible for these collective behaviors.

### 6.2. Different Approaches to Analyze the Intra-S Phase Checkpoint Action on the Replication Program

Few theoretical studies have focused on the effects of stalled or slowed forks and checkpoint activation on the replication program. In an initial work [184], the authors tried to address the dynamics of dormant origin usage in a replicon cluster based on DNA combing data in the absence or presence of HU in human cells. They simulated a circular DNA template of 250 kb with a variable number of licensed origins and a log-normal distribution of initiation probability for the origins in the cluster. They showed that an excess number of licensed origins can, per se, protect against replication failure due to stalled forks, independently from the probability of initiation of the origins.

Gauthier and colleagues [185] used a rate-equation approach to study the DNA replication kinetics in the presence of replication defects based on inter-origin distances data from budding yeast, the *Xenopus* in vitro system, and human cells. In the limit of a very long repair time (τ → ∞), they find a crossover between two regimes: a normal regime, where the influence of defects is local and does not affect the average inter-origin distances, and an initiation-limited regime, where the number of initiation events required to complete replication increases drastically. They showed that in the majority of eukaryotes, the crossover between the two regimes occurs when the density of defects is equal to the density of fired origins during an unchallenged S phase. Using this interesting finding, we can make the following thought experiment: assuming that an organism lacking checkpoint mechanisms is facing extensive replication stress, its replication dynamics follow an initiation-limited regime. This regime requires an important amount of replication factors, that might be unavailable, to complete S phase. Therefore, applying this thought experiment to a real organism, the crossover could be considered as a threshold for the activation of the intra-S phase checkpoint that blocks late origin firing, thereby increasing the amount of replication factors and so allowing more origins to fire close to stalled forks.

The action of Chk1 on the replication program was analyzed both experimentally and theoretically in the *Xenopus* in vitro system [69]. Based on I(t) data from DNA combing experiments in the presence or absence of a specific Chk1 inhibitor (UCN-01), the authors proposed a numerical model where Chk1 inhibits origin firing everywhere along the genome with a certain probability, while an unknown factor, a possible candidate being Plk1, blocks the inhibitory action of Chk1 in early active clusters as shown in Figure 1c. This model reproduces accurately the I(t), meaning the temporal program of origin firing (Figure 4a,b), but cannot reproduce the experimental inter-origin distances distribution, the spatial program of origin firing (Figure 4c). Therefore, further analysis is necessary to understand how the intra-S phase checkpoint acts to regulate both the spatial and temporal program of origin firing.

## 7. Conclusions

Many advances have been made in deciphering the molecular mechanism of the initiation of DNA replication and describing the spatio-temporal replication program with regards to the intra-S phase checkpoints. Further experimental and theoretical analysis is necessary in order to confirm and distinguish between the proposed models, especially in multicellular organisms. Recent advances in high throughput analysis of single molecules (e.g., nanopore sequencing) will tell how and where the intra-S phase checkpoints modulate the replication program on single DNA molecules in a population of replicating cells in order to maintain genome stability during development and differentiation. Numerical models will therefore be valuable tools to analyze these high throughput data and to better understand the interplay between the fundamental processes of genome duplication and cell cycle control.

## Figures and Tables

**Figure 1 genes-10-00094-f001:**
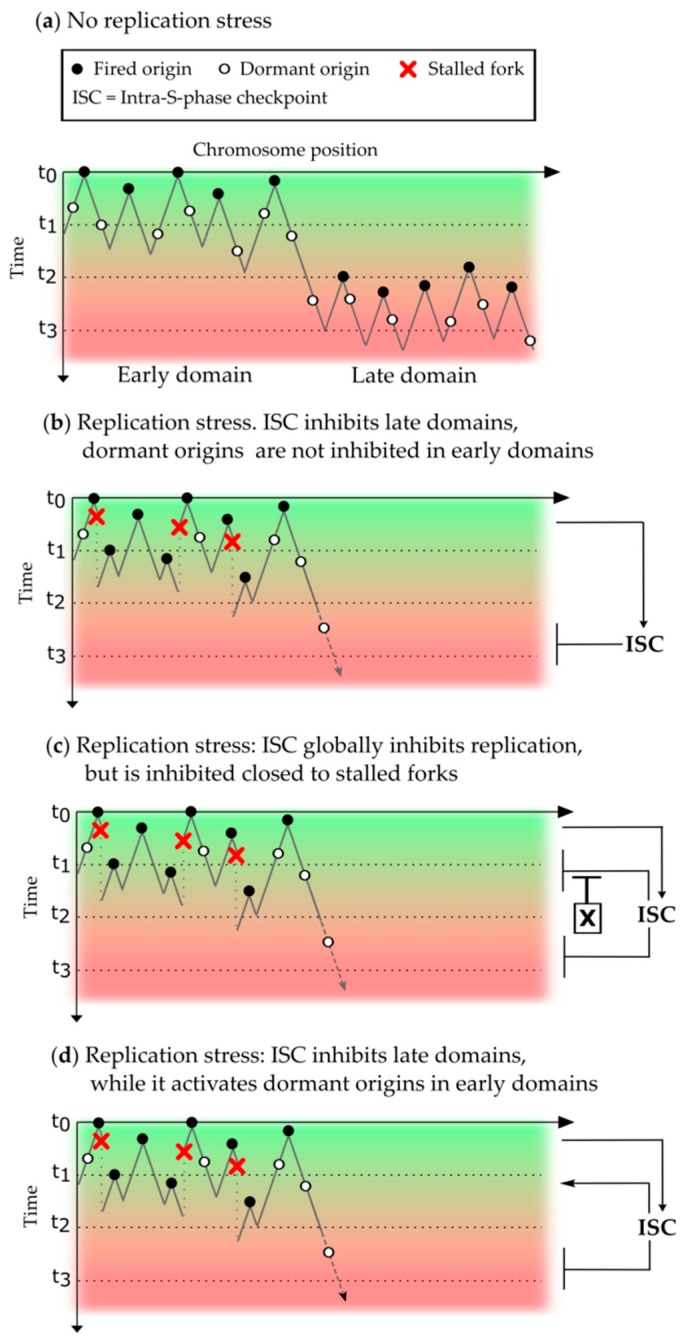
Models of the regulation of the spatio-temporal replication program by the intra-S phase checkpoint. The “*y*-axis” represents the S phase time, where t_0_ is the start of S phase and t_1_ < t_2_ < t_3_ are three subsequent arbitrary times smaller than the duration of S phase. The “*x*-axis” is the chromosomic position. Black lines symbolize replicated DNA, and valleys represent terminations. (**a**) In the absence of replication stress, replication origins fire in early domains (green region) or late domains (red region). In the presence of replication stress, the intra S phase checkpoint (ISC) inhibits origin activation in late domains and three different models are proposed to explain the firing of dormant origins inside early domains: (**b**) The ISC passively allows dormant origins firing close to stalled forks. Due to a slight asynchrony between early and dormant origins in early domains, pre-initiation complexes (preICs) on dormant origins might have passed the ISC inhibited steps, (**c**) ISC action is inhibited inside early domains by an unknown mechanism X, maybe by Plk1 interaction with MCM2-7 [71] closed to stalled forks, and (**d**) ISC locally activates dormant origins inside early domains by an unknown mechanism.

**Figure 2 genes-10-00094-f002:**
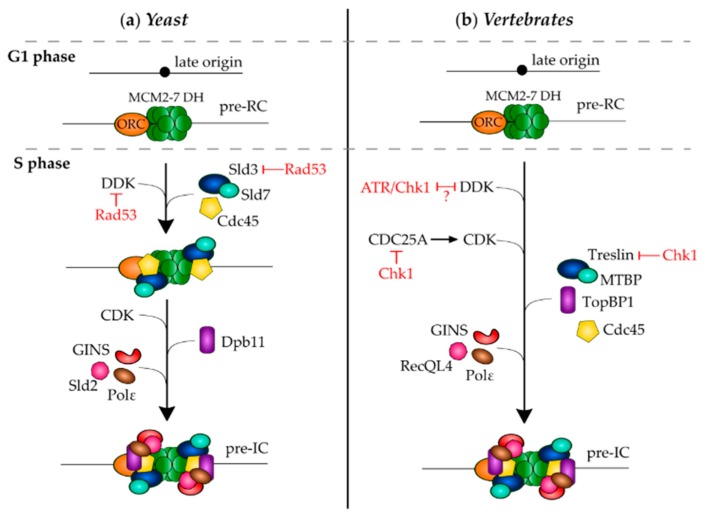
Mechanisms for inhibition of preIC assembly on late origins by the intra-S phase checkpoint in (**a**) budding yeast and (**b**) vertebrates. Rad53 phosphorylates and inhibits Dbf4 and Sld3 in budding yeast. In vertebrates, Chk1 inhibits Treslin mediated Cdc45 recruitment and Cdc25 mediated CDK activation. Inhibition of DDK activity by ATR/Chk1 is unclear (see text).

**Figure 3 genes-10-00094-f003:**
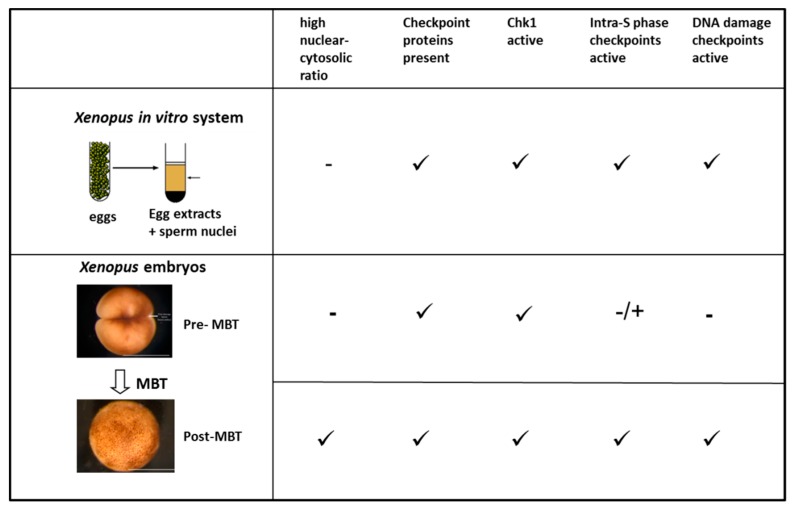
Comparison of checkpoint responses in the *Xenopus* in vitro system, in pre- and post-mid-blastula transition (MBT) *Xenopus* embryos. Photos of *Xenopus* embryos are from Xenbase [166].

**Figure 4 genes-10-00094-f004:**
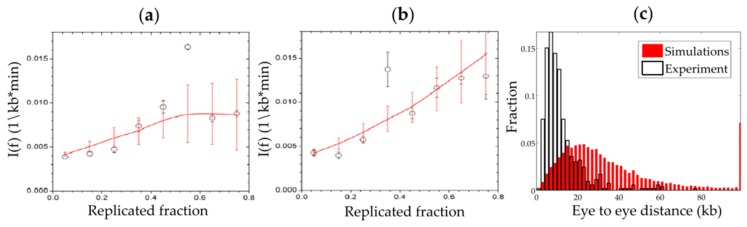
Numerical simulation of the intra-S phase checkpoint action reproduces the temporal but not the spatial program of origin firing. The frequency of origin initiation and the replication of eye-to-eye distance distribution were extracted from DNA combing experiments in the absence or presence of Chk1 activity. The model discussed in the text was simulated by using a Monte Carlo technique. The plots represent the experimental (black) and numerical (red) frequency of initiation in (**a**) the presence or (**b**) the absence of Chk1 activity from [69], and (**c**) the eye-to-eye distance distribution in the presence of Chk1 activity (unpublished data).

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
