# Peer review of "On the Interplay of the DNA Replication Program and the Intra-S Phase Checkpoint Pathway"

_genes, 2019, doi:10.3390/genes10020094_

Round 1

Reviewer 1 Report

This review makes a good contribution to the current knowledge around DNA replication program and its regulation by checkpoint control. I would like to request the authors to address the following concerns:

Lines 136 to 145: The authors have noted a good point that all replication stress may not activate ATR to significantly high levels. Technical limitations to be able to obtain a localized resolution to detect and analyze low levels of DNA replication stress in cells makes it challenging. They also suggest measurement of origin firing and replication fork progression as clearer alternatives, I believe they are suggesting DNA fiber technique, in which case, it is a great technique to analyze DNA replication and also commonly used these days but the limitation being protein localization to the sites of replication stress can’t be addressed in these assays. They can elaborate on why they think measuring replication progression is a better choice and how can one or a few stalled forks be identified among the globally replicating genome in these assays?

Author Response

Point 1:

Lines 136 to 145: The authors have noted a good point that all replication stress may not activate ATR to significantly high levels. Technical limitations to be able to obtain a localized resolution to detect and analyze low levels of DNA replication stress in cells makes it challenging. They also suggest measurement of origin firing and replication fork progression as clearer alternatives, I believe they are suggesting DNA fiber technique, in which case, it is a great technique to analyze DNA replication and also commonly used these days but the limitation being protein localization to the sites of replication stress can’t be addressed in these assays. They can elaborate on why they think measuring replication progression is a better choice and how can one or a few stalled forks be identified among the globally replicating genome in these assays?

Response 1

We think that indeed measuring low RS via small changes in fork densities or origin distances using techniques like DNA combing is a very sensitive method. But the referee is right pointing out that this technique does not allow localizing the RS to specific sequences in the genome. We actually thought there about another technique which has just been developed using nanopore-sequencing (2 studies from the Hyrien and Nieduszynski group on bioRxiv) applied to replication studies which could in future allow directly detecting and localising stalled forks on the genome on the level of single DNA fibers.

We propose to add in the text at the end of paragraph (including two new references which modified the reference list):

 We therefore believe that the clearest read-out of RS is the direct measurement of replication origin firing and replication fork progression by sensitive methods like DNA fiber stretching techniques such as DNA combing. This technique allows visualizing the replication process at the level of single DNA fibers, but the throughput of such a technique is low if one needs to map the genomic position of the observed fiber. Very recent advances in the study of DNA replication studies using nanopore sequencing [38,39] could soon allow directly detecting few stalled forks at genomic positions on single DNA fibers with high throughput.

Point 2 : English language and style are fine/minor spell check required

Response 2

The manuscript has now been read and corrected by a native English speaker (see acknowledgements)

Reviewer 2 Report

The review discusses important aspects of Intra-S checkpoint. The review goes in great detail about the literature on intra-S checkpoints in eukaryotes, and is a good attempt at providing a comprehensive set of information. 

While the authors do discuss a lot of significant discoveries made in the field, at places, many findings are listed without drawing appropriate conclusions from these findings as an ensemble. For example: In lines 212-233, the authors need to use the listed findings and present it more logically and coherently, for the reader to follow the subsequent inferences. With the current premise, the 3 scenarios of IS checkpoint and origin firing seems confusing and can be explained better in the text (figure 1).

The abstract must contain some more details on the models discussed in the paper and their significance. The current information in the abstract seems very vague and the purpose of the review does not become clear until the last quarter of the manuscript.

Since, the authors discuss Xenopus related S-phase studies in detail, it seems logical that they include the findings from Dr. JC Walters about inter-strand crosslinks, replication fork stalling and Fanconi Anemia proteins. While the authors do refer to FANCI, they exclude many other findings on FA proteins and checkpoint activation.

Finally, the conclusion lacks a definite perspective. Like the abstract, the perspective provided is very brief and could be elaborated to explain the authors’ vision for future research.

Minor Comments

-          When referring to Xenopus system in the text, please mention clearly if you refer to experiments done in Xenopus embryos or in Xenopus S-phase replicating egg extract system. The system is explained in greater detail in the second half of the paper, but it will help to have some clarity at the onset itself.

-          Some of the English used has grammatical errors making the message not easily comprehensible. Please have the article proofread for language edits.  While using quotation marks, there is a discrepancy in the opening quotes punctuation. Please ensure the right font/punctuation is used.

Author Response

Point 1

While the authors do discuss a lot of significant discoveries made in the field, at places, many findings are listed without drawing appropriate conclusions from these findings as an ensemble. For example: In lines 212-233, the authors need to use the listed findings and present it more logically and coherently, for the reader to follow the subsequent inferences.

Response 1

We propose to re-order the paragraph now in separating studies in mammalian cells and Xenopus, respecting the chronological order of the studies within. We also added another conclusion at the end of this paragraph (in bold), in addition to the ones already in the text (beginning of this paragraph, start of the next paragraph).

In mammalian or avian cell lines and in the Xenopus in vitro system, initiation of DNA replication was shown to be regulated by the intra-S phase checkpoint at different levels: single replication origins, clusters, domains and replication foci. In mammalian cells, origins are inhibited at the ribosomal DNA (rDNA) locus in cells proficient for the checkpoint pathway after ionizing radiation, but not in AT cells [61]. The intra-S checkpoint inhibits the DNA synthesis in late replication foci in the presence of the replication inhibitor aphidicolin [47,62]. In asynchronous human and avian cells, ATR/ATM inhibition by caffeine, or Chk1 inhibition by UCN-01 or by Chk1 depletion, led to an increased origin density inside already activated clusters in all tested cell lines, and an increase in foci number only in avian cell lines [63].  In human cells, genome wide origin mapping on coding sequences by NGS showed that upon fork stalling in HU, only early origins fire in early domains, but upon checkpoint inhibition with the ATM/ATR inhibitor caffeine, origins fire in mid-to late regions [64]. In the embryonic Xenopus in vitro system using single DNA fiber analysis, it was shown that ATR regulates origin firing during an unchallenged S phase [65,66], but no effect of Chk1 depletion or inhibition was detected in the Xenopus in vitro system [67,68]. Later, two other studies clearly showed that Chk1 depletion or inhibition in the Xenopus in vitro system resulted in increased  overall origin firing [69,70], demonstrating the role of Chk1 in the negative regulation of DNA replication in the absence of exogenous replication stress. Upon Chk1 or ATR inhibition an important increase of global fork densities was found, but no difference in local origin distances was observed after Chk1 inhibition and only a modest decrease was seen upon caffeine treatment [65,70]. These two apparently contradictory results suggest that the checkpoint mainly inhibits origins at the level of later replicating clusters in the Xenopus in vitro system, but less inside already activated clusters, in contrast to mammalian cell lines. This difference between differentiated cells and the non-differentiated embryonic system might be linked to the overall higher fork density and therefore less available dormant origins in the shorter embryonic S phase.

A general conclusion of the yeast and mammalian/Xenopus systems paragraphs was stated at the beginning of the next paragraph. In order to make this more obvious we conclude now:

These different observations illustrate that the checkpoint inhibits origin firing at different chromosomal organization levels and often in inactive, late clusters or domains, depending on the organisms.

Point 2

With the current premise, the 3 scenarios of IS checkpoint and origin firing seems confusing and can be explained better in the text (figure 1).

Response 2

We changed the caption of Figure 1, added missing Figure 1 a-d citations in the main text and explained in more detail in the main text:

Caption of Figure 1 :

Models of the regulation of the spatio-temporal replication program by the intra-S-phase checkpoint. The “y-axis” represents the S-phase time, where t0 is the start of S phase and t1<t2<t3 are three subsequent arbitrary times smaller than the duration of S phase. The “x-axis” is the chromosomic position. Black lines symbolize replicated DNA, and valleys represent terminations. (a) In the absence of replication stress, replication origins fire in early domains (green region) or late domains (red region)., In the presence of replication stress, the ISC inhibits origin activation in late domains and three different models are proposed to explain the firing of dormant origins inside early domains: (b) The ISC passively allows dormant origins firing close to stalled forks: Due to a slight asynchrony between early and dormant origins in early domains, preICs on dormant origins might have passed the ISC inhibited steps, (c)  ISC action is inhibited inside early domains by an unknown mechanism X, maybe by Plk1 interaction with MCM 2-7 [72] closed to stalled forks, (d) ISC locally activates dormant origins inside early domains by an unknown mechanism.

We changed the text of this paragraph :

In order to explain how the checkpoint in response to DNA damage can simultaneously inhibit and promote origin firing, different models have been proposed [71] which could also be valid for the regulation in response to endogenous or exogenous replication stress (Figure 1). In the absence of replication stress, origins in early domains are activated in the beginning of S phase (on the time scale t0-t1) before origins in late domains at consecutive times (t2-t3) later in S phase (Figure 1a). In the presence of replication stress, the activation of origins in late domains is inhibited by the intra-S phase checkpoint (ISC) (Figure 1 b-d). Since the inhibitory checkpoint signal is produced in early domains where replication forks are stalled, it should normally inhibit origins close to stalled forks, before it is transmitted to late domains. There could be at least three, not mutually exclusive possibilities, to explain dormant origin firing in these early domains. First, dormant origins can fire because they have already assembled functional preICs, due to only a slight asynchrony in the assembly of the preICs between early and dormant origins (Figure 1b). Thus, the inhibitory step of the checkpoint in the activation cascade of origins has been passed. In a second possibility, there could be a local inhibition of the inhibitory action of the checkpoint close to stalled forks inside early domains, resulting in the activation of dormant origins (Figure 1c). This unknown mechanism (X) could involve for example Plk1, recruited in a checkpoint dependent manner via MCM2-7 to stalled forks and leading to a decrease of Chk1 activity [72]. It could also implicate other checkpoint recovery mechanisms like phosphatases directly inhibiting Chk1 (see chapter 4). Third, the checkpoint could directly activate dormant origins inside early clusters by targeting proteins that are distinct from those responsible for late origin inhibition. Future studies will help to distinguish between these scenarios or reveal others.

Point 3

The abstract must contain some more details on the models discussed in the paper and their significance. The current information in the abstract seems very vague and the purpose of the review does not become clear until the last quarter of the manuscript.

Response 3

We changed the abstract by adding one sentence (in bold) about the model and shortening elsewhere. The length for the abstract is limited to 200 words which does not allow to go into more detail. The purpose of the review is also stated at the end of the introduction.

DNA replication in eukaryotes is achieved by the activation of multiple replication origins which needs to be precisely coordinated in space and time. This spatio-temporal replication program is regulated by many factors to maintain genome stability, which is frequently threatened through stresses of exogenous or endogenous origin. Intra-S phase checkpoints monitor the integrity of DNA synthesis and are activated when replication forks are stalled. Their activation leads to the stabilization of forks, to the delay of the replication program by the inhibition of late firing origins and the delay of G2/M phase entry. In some cell cycles during early development these mechanisms are less efficient in order to allow rapid cell divisions. In this article, we will review our current knowledge on how the intra-S phase checkpoint regulates the replication program in budding yeast and metazoan models, including early embryos with rapid S phases. We sum up current models on how the checkpoint can inhibit origin firing in some genomic regions, but allows dormant origin activation in other regions. Finally, we discuss how numerical and theoretical models can be used to connect the multiple different actors into a global process and to extract general rules.

Point 4

Since, the authors discuss Xenopus related S-phase studies in detail, it seems logical that they include the findings from Dr. JC Walters about inter-strand crosslinks, replication fork stalling and Fanconi Anemia proteins. While the authors do refer to FANCI, they exclude many other findings on FA proteins and checkpoint activation.

Response 4

As the referee points out, FA proteins play a central role in the replication coupled repair of interstrand-cross links which is not the scope of our review.  We wanted to mention a novel role the role of FANC proteins in response to replication stress (HU) connected with origin activation and not to yet another role in promoting fork stability.  We cited one paper which describes the positive role of one FANC protein, FANCI, in the activation of dormant origins, which we thought to be most relevant for the comprehension of the different models in Figure 1. Few other studies, to our knowledge, describe that in the absence of FANCD2 dormant origin firing increases. We propose to cite one of many recent relevant FA reviews (Michl et al 2016, EMBOJ) in addition, and add the citations on FANCD2 to modify our text:

Fanconi anemia (FA) pathway is known to repair DNA interstrand cross-links during S phase by orchestrating the interplay between multiple DNA repair pathways [115,116]. In response to low replication stress (HU) and in the absence of FANCD2 and FANCI, core components of the FA complex, origin firing is altered. FANCD2 and FANCI interact with the MCM2-7 complex at the replisome [117] and are involved in the regulation of dormant origin firing [118–120].

Point 5

Finally, the conclusion lacks a definite perspective. Like the abstract, the perspective provided is very brief and could be elaborated to explain the authors’ vision for future research.

Response 5

We made a conclusion rather than a perspective. However, we can sum up more points discussed in the main text and propose to add in bold:

Many advances have been made to decipher the molecular mechanism of the initiation of DNA replication and to describe the spatio-temporal replication program with regards to the intra-S phase checkpoints. Further experimental and theoretical analysis is necessary in order to confirm and distinguish the proposed models, especially in multicellular organisms.  Recent technological advances in high throughput analysis of single molecules (for exemple nanopore sequencing) will tell how and where the intra-S phase checkpoints modulate the replication program on single DNA molecules in a population of replicating cells in order to maintain genome stability during development and differentiation. Numerical models are therefore valuable tools to analyze these high throughput data and to better understand the interplay between the fundamental processes of genome duplication and cell cycle control

Minor points

Point 6

When referring to Xenopus system in the text, please mention clearly if you refer to experiments done in Xenopus embryos or in Xenopus S-phase replicating egg extract system. The system is explained in greater detail in the second half of the paper, but it will help to have some clarity at the onset itself.

Response 6

We now mention the Xenopus in vitro system and Xenopus embryos at the end of the introduction. For more clearity, we also replace Xenopus embryos by the longer term Xenopus in vitro system where we think it is necessary, although it is not really current in the field since the Xenopus in vitro system “represents” early Xenopus embryos ( 1. S phase after fertilisation). But we point out the differences between in vivo and in vitro in the early development chapter.

Point 7

Some of the English used has grammatical errors making the message not easily comprehensible. Please have the article proofread for language edits.

Response  7

The manuscript has now been proofread by a native English speaker.

Point 8

While using quotation marks, there is a discrepancy in the opening quotes punctuation. Please ensure the right font/punctuation is used.

Response  8

The font/punctuation has been corrected.

Reviewer 3 Report

DNA replication in eukaryotes is a highly dynamic process. A fully-assembled eukaryotic replisome is a multiprotein machine that contains over two dozen proteins, including DNA polymerases, sliding clamps, replicative helicase, and primase along with several factors that participate in cell cycle and checkpoint control. In this review, the authors describe the intra-S phase checkpoint pathway on this highly regulated DNA replication process. This review is integrated and well-written review.  

Major points:

1. In Figure 1, can authors give more explanation or definition about four time points (t0 – t3) and chromosome position? I did not find any descriptions of these two in the figure legend and main text. 

2. In Figure 2a and 2b, according to current model of DNA replication initiation, two Mcm2-7s are loaded to one Orc complex and form an NTD-to-NTD double hexamer. I found authors had mentioned “head-to-head DH” in the text, but in these two panels, readers may confuse it as one Mcm2-7. Can authors specifically emphasize there are two Mcm2-7s here?

Minor points:

1. Page 13, line 531, what is the meaning of KJMA, I did not find it in the abbreviation list.

Author Response

Point 1

1. In Figure 1, can authors give more explanation or definition about four time points (t0 – t3) and chromosome position? I did not find any descriptions of these two in the figure legend and main text. 

Response 1

We added the following explanation in the caption of Figure 1 for a better understanding:

Models for the regulation of the spatio-temporal replication program by the intra-S-phase checkpoint.  The “y-axis” represents the S-phase time, where t0 is the start of S phase and t1<t2<t3 are three subsequent arbitrary times smaller than the duration of S phase. The “x-axis” is the chromosomic position. Black lines symbolize replicated DNA, and valleys represent terminations. (1) In the absence of replication stress….

In the main text we added (lines 311-13):

:

In the absence of replication stress, origins in early domains are activated in the beginning of S phase (on the time scale t0-t1) before origins in late domains at consecutive times (t2-t3) later in S phase (Figure 1a).

Point 2

2. In Figure 2a and 2b, according to current model of DNA replication initiation, two Mcm2-7s are loaded to one Orc complex and form an NTD-to-NTD double hexamer. I found authors had mentioned “head-to-head DH” in the text, but in these two panels, readers may confuse it as one Mcm2-7. Can authors specifically emphasize there are two Mcm2-7s here?

Response 2

We modified the Figure 2a, b by clearly separating the two MCM rings with a black line rather than a white shadow which is misleading. We also replaced MCM2-7 by MCM2-7 DH. Please see uploaded corrected Figure 2. The corrected figure has been inserted into the revised manuscript.

Point 3

Page 13, line 531, what is the meaning of KJMA, I did not find it in the abbreviation list.

Response 3

KJMA (Kolmogorov-Johnson-Mehl-Avrami) has been added to the abbreviation list. These are the authors at the origin of the KJMA model developed in the 1930s to explain the kinetics of crystallization.

Point 4 Extensive editing of English language and style required

Response 4

The manuscript has now been read and corrected by a native English speaker (see acknowledgments).

Reviewer 4 Report

On the interplay of the DNA replication program and the intra-S phase checkpoint pathway

By Diletta Ciardo , Arach Goldar  and Kathrin Marheineke

This review article summarizes recent work on how DNA replication checkpoints are regulated in metazoans. The authors then discuss some numerical simulations and theoretical models for regulation of the events that initiate DNA replication and how the intra-S phase checkpoint would act to regulate these events.

This article is well-written and I have no substantive changes to suggest. 

I would only suggest:

The addition of some commas, such as before ‘whereas,’ before and after ‘therefore,’ after subordinate clauses, etc. In addition, ‘funded’ on line 531 is probably meant to be ‘founded.’

Author Response

Point 1

The addition of some commas, such as before ‘whereas,’ before and after ‘therefore,’ after subordinate clauses, etc. In addition, ‘funded’ on line 531 is probably meant to be ‘founded.’

Response 1 :

These specific errors have been corrected. In addition, the manuscript has been read and corrected by a native English speaker (see acknowledgments).